# Effects of Late Gestation Supplements Differing in Fatty Acid Amount and Profile to Beef Cows on Cow Performance, Steer Progeny Growth Performance through Weaning, and Relative mRNA Expression of Genes Associated with Muscle and Adipose Tissue Development

**DOI:** 10.3390/ani13030437

**Published:** 2023-01-27

**Authors:** Taoqi Shao, Joshua C. McCann, Daniel W. Shike

**Affiliations:** Department of Animal Sciences, University of Illinois at Urbana-Champaign, Urbana, IL 61801, USA

**Keywords:** beef cows, fatty acids, fetal programming, late gestation, mRNA expression

## Abstract

**Simple Summary:**

Nutrition during late gestation has the potential to alter future progeny performance. Fatty acids are important for the regulation of protein and lipid metabolism. We investigated the effects of different fatty acid supplements during late gestation on cow performance, steer progeny growth performance, and gene expression through the weaning and backgrounding phases. The results indicate that late gestation fatty acid supplementation modifies plasma fatty acid concentration and alters muscle and adipose gene expression. However, the fatty acid supplements evaluated did not impact cow performance or steer growth performance through weaning and backgrounding.

**Abstract:**

Strategic supplementation during late gestation has the potential to alter progeny performance. Mature fall-calving Simmental × Angus cows were used to evaluate the effects of late gestation supplementation of fatty acids to beef cows on cow performance, steer progeny growth performance during pre-weaning and backgrounding periods, and relative mRNA expression of genes associated with myogenesis and adipogenesis. Cows (n = 190; 4 pasture groups of cows/treatment) grazed endophyte-infected tall fescue and were supplemented during late gestation with calcium salts of either saturated fatty acid/monounsaturated fatty acid (SFA/MUFA), polyunsaturated fatty acid (PUFA), or an isocaloric and isonitrogenous control (CON). There were no differences (*p* ≥ 0.11) in cow body weight (BW) or body condition scores from pre-supplementation to weaning or steer BW at birth, weaning, or at the end of the backgrounding period. Concentrations of C18:2n-6 in plasma were greater (*p* = 0.01) in SFA/MUFA and PUFA cows compared to CON cows during supplementation. For mRNA expression in the longissimus muscle of steer progeny from birth to weaning: PAX7 decreased to a greater (*p* < 0.01) extent for SFA/MUFA and PUFA steers; AGPAT1 and CPT1 increased to a greater (*p* ≤ 0.02) extent for CON steers. The expression of MYH7 mRNA during the pre-weaning period was greater (*p* = 0.01) in PUFA. In conclusion, late gestation fatty acid supplementation modified plasma relative concentrations of fatty acids for dams and progeny and modified mRNA expression of genes related to myogenesis and adipogenesis but had limited effects on progeny growth performance during pre-weaning and backgrounding periods.

## 1. Introduction

Fetal programming is the response to a specific challenge to the organism during a critical development period that leads to persistent effects [1]. Because the majority of muscle fibers are formed during the fetal stage, when skeletal muscle has a lower priority in nutrient partitioning, muscle development is more vulnerable to nutrient deficiency [2]. Nutritional deficiency during late gestation reduces the density of satellite cells, which could also negatively affect postnatal muscle growth [3]. The sequential adipogenesis in different depots provides an opportunity to specifically enhance intramuscular adipocyte formation during the late gestation, neonatal, and early weaning stages [4]. Nutritional manipulations, such as maternal supplementation of nutrients or bioactive compounds, during these time windows are expected to be able to modify intramuscular adipogenesis [4,5]. Marbling is correlated with the number and size of intramuscular adipocytes [5]. Upregulated adipogenesis in intramuscular fat during the fetal stage leads to an increased number of intramuscular adipocytes, which could accumulate lipids and form marbling during postnatal growth [5]. Therefore, proper fetal programming is significant for achieving the maximum growth and production potential of animals.

Fatty acids are important ligands for the regulation of protein and lipid metabolism. The function of peroxisome proliferator-activated receptor gamma (PPARG), which is the key gene for adipogenic differentiation [4], is stimulated by n-6 fatty acids, such as C18:2 [6,7]. Conversely, n-3 fatty acids, such as C20:5n-3 (EPA) and C22:6n-3 (DHA), can decrease the transcription of lipogenic genes and increase the transcription of lipolytic genes by activating PPARA. However, the effects of EPA and DHA on fetal muscle development have been inconsistent. Studies that used the C2C12 myotube model indicated that EPA and DHA promote muscle protein synthesis and improve insulin resistance in muscles [8,9]. It was also reported that overdosed EPA and DHA inhibited C2C12 myoblasts’ proliferation and differentiation, as well as downregulated muscle-related gene expression [10,11]. Previous studies [12,13] have indicated that supplementing polyunsaturated fatty acids to dams during late gestation did not impact offspring pre-weaning growth or health performance, but improved BW gain during the finishing phase. Conversely, Shao et al. [14] reported decreased steer weaning BW when fall-calving dams grazing endophyte-infected tall fescues were supplemented with a similar ratio of polyunsaturated fatty acid supplements as in Marques et al. [13]. Not only the quantities but also the balance between n-6 and n-3 fatty acids is important for animal health and growth performance [15]. However, the amount and ratio of n-6 to n-3 that would improve beef production, as well as the mechanisms, are still yet to be determined. Since unanticipated poorer pre-weaning performance was observed in calves born from dams supplemented with polyunsaturated fatty acids in a previous study [14], the current study increased the inclusion of n-6 enriched Ca salts of fatty acids in the polyunsaturated fatty acid treatment, as well as added a no-added-fat control supplement to compare non-fat supplementation to fatty acid supplementation. The objective of the current study was to investigate the effects of fatty acid supplementation and fatty acid profile differences during late gestation in beef cows on cow performance, steer progeny growth performance during pre-weaning and backgrounding periods, and relative mRNA expression of genes relative to myogenesis and adipogenesis at birth and weaning.

## 2. Materials and Methods

Experimental animals were managed according to the guidelines recommended in the Guide for the Care and Use of Agriculture Animals in Agriculture Research and Teaching (Federation of Animal Science Societies, 2010). All experimental procedures followed were approved by the University of Illinois Institutional Animal Care and Use Committee (IACUC Protocol #17292).

### 2.1. Experimental Design, Animals, and Diets

One hundred and ninety fall-calving Angus × Simmental cows (initial BW = 612 ± 69 kg, BCS = 6 ± 1) from Dixon Springs Agricultural Center, Simpson, IL, USA were utilized for the current study. Prior to supplementation (d -82), cows were stratified by BW, BCS, and age and then randomly assigned to 12 grazing groups with 15–16 cows/pasture. Cow grazing groups were randomly assigned to 1 of the 3 treatments for the last 82 ± 5 d of gestation: 0.16 kg DM/cow/d soybean hulls mixed with 0.91 kg DM/cow/d whole-shelled corn (CON), 0.77 kg DM/cow/d soybean hulls mixed with 0.155 kg DM/cow/d EnerGII (SFA/MUFA, rich in palmitic and oleic acids), or 0.77 kg DM/cow/d soybean hulls mixed with 0.12 kg DM/cow/d Prequel and 0.04 g DM/cow/d Strata (PUFA, rich in linoleic acid, eicosapentaenoic acid, and docosahexaenoic acid). EnerGII, Prequel, and Strata (Virtus Nutrition, LLC, Corcoran, CA, USA) are calcium salts of fatty acids. Cows were grazing on predominantly endophyte-infected tall fescue pastures with an average stocking rate of 1.6 cows per hectare during the supplementation period and rotated as needed. Forage availability during the supplementation period was visually evaluated by trained personnel. Supplement composition and nutrient intake information are presented in Table 1. The fatty acid intake in Table 1 was calculated according to the amount fed to the group and the number of cows in each group. The three supplements were designed to be isocaloric and isonitrogenous. The nutritional and fatty acid profiles of late gestation dietary ingredients are presented in Table 2. Supplementation started on 6/28/2019. Cows were supplemented every Monday, Wednesday, and Friday in 4 portable bunks (4.88 × 0.76 m, accessible from both sides) for each grazing group. The average bunk space was 1.4 to 1.5 m per cow. Typically, supplements were ingested by cows within 10–15 min. Weather data were collected from the Water and Atmospheric Resources Monitoring Program (Illinois Climate Network & Illinois State Water Survey; Champaign, IL, USA).

Cows were weighed and body condition scored (1–9 scale) at the initiation of the supplementation (d -83 and -82), middle of supplementation (d -42), within one week post-calving (4 ± 2 d post-calving), at weigh-suckle-weigh (WSW; d 68); at AI (d 85), AI-pregnancy check (d 120), and at weaning (d 174) to monitor cow performance. Once weekly, cows that had calved and their calves were pulled out from the treatment groups and comingled to a common pasture, and were managed the same. Cows that aborted, had a stillborn calf, or had a heifer calf were removed from the trial immediately after being detected. In order to decrease variation in growth performance and mRNA expression, only cows with steer calves (97 pairs) were utilized for post-calving investigation. After comingling to the common pasture, cow/calf pairs were provided with 2.27 kg/cow/d of dried distiller grains with solubles (DDGS; 88% DM, 30% CP, 7% either extract, 39% NDF, 11% ADF) and soybean hulls (84% DM, 11% CP, 0.8% either extract, 64% NDF, 45% ADF) in a ratio of 50:50. Cattle were also provided with ad libitum hay (80% DM, 7% CP, 71% NDF, and 39% ADF) from d 20 postpartum to weaning as forage availability declined in the fall. Cows were synchronized using the 7-day Co-Synch + controlled internal drug-release (CIDR; Pfizer Animal Health, New York, NY, USA) procedure [16] and artificially inseminated (85 ± 5 d post-calving) with sexed heifer semen from a single sire. Ten days after AI, cows were exposed to two clean-up bulls that had passed the breeding soundness examination for 62 d. Pregnancy diagnoses were performed on d 120 and d 189 by a trained technician with ultrasonography (Aloka 500 instrument, Wallingford, CT, USA).

Cows were vaccinated at the initiation of supplementation (d -83) and the middle of the supplementation (d -42). On d -83, 2 mL Leptoferm-5 (Zoetis, Florham Park, NJ, USA), 1 mL Anaplasmosis vaccine (University Products L.L.C., Baton Rouge, LA), and 2 mL Auto. M. Bovis. (Pinkeye vaccine customized by Newport Laboratories, Worthington, MN, USA) was administered to the experimental cows. In addition, 2 Patriot Insecticide Cattle Ear Tags (Bayer, Shawnee Mission, KS, USA) and Ivermectin (Norbrook, Newry, UK) were applied. On d -42, 5 mL Bovishield Gold FP5 VL5 HB (Zoetis), 5 mL Covexin 8 (Merck Animal Health, Rahway, NJ, USA), 7 mL MU-SE (Zoetis), and 2 mL ScourGuard 4KC (Zoetis) and Cylence (Bayer) were administered.

From calving season to weaning, there were 6 pairs removed. One pair from CON and two pairs from PUFA were removed at weigh-suckle-weigh due to the loss of calf by predators. Two pairs from CON were removed at weigh-suckle-weigh because of cow mastitis and poor calf conditions. One pair from PUFA was removed after AI because the cow had lymphoma.

Calf birth BW was recorded, and bull calves were castrated within 24 h after calving. The vaccination program for calves during the pre-weaning period was similar to the program reported by Shao et al. [14]. The only difference was the dates of the vaccines being at birth, d 68, and d 173. Steers were weighed and weaned at 174 ± 5 d of age. Thereafter, the ninety-one steer calves were fed a backgrounding diet (Table 3) in concrete bunks for 42 d. Pre-weaning and backgrounding performance were evaluated based on BW and ADG.

### 2.2. Sampling

Blood samples were collected from cows at pre- (d -82) and mid-supplementation (d -40), and from cow/steer pairs within one week after calving (4 ± 2 d post-calving). Blood samples (10 mL) were collected from the jugular vein of cows and calves by using polypropylene tubes (BD Vacutainer) containing sodium heparin for plasma, and placed on ice. After centrifugation for 20 min at 2000× *g* and 4 °C, plasma was stored at −80 °C until later analysis. 

Milk production was determined before breeding (67 ± 5 d postpartum) on a subset of cows via WSW, as described by Beal et al. [17]. Cow BW at calving, cow age, and day postpartum were similar across subset groups (4–8 pairs/group). Milk samples were collected by hand stripping on a random subset of cows (4 cows/group) for milk composition and fatty acid profile analysis. Milk samples for composition analysis were placed in a cooler with ice packs underneath the samples and shipped to Dairy Lab Service Inc. (Dubuque, IA, USA). Milk samples for fatty acid profile analysis were placed in a cooler with dry ice underneath the samples and shipped to Cumberland Valley Analytical Service Inc. (Waynesboro, PA, USA).

Feed samples for each ingredient, including pasture forage and supplements, were collected every 2 weeks during supplementation and lactation periods for proximate and fatty acid profile analysis. All feed samples were stored in −20 °C until further processing.

Longissimus muscle and subcutaneous adipose tissue biopsy samples for relative mRNA expression analysis were collected from every steer calf at birth (4 ± 2 d of age) and from a subset of steers (n = 48; 4 steers from each grazing group) at 3 wk prior to weaning (d 154). The subset of steers was selected for pre-weaning biopsy based on their BW on d 120 being representative of the group average. Biopsy procedures were previously described by Shao et al. [14].

### 2.3. Analytical Procedures

Thawing and pooling procedures for plasma samples were conducted using the methods described previously by Shao et al. [14]. Relative concentrations of fatty acids in pooled plasma samples were analyzed by the Metabolomics Center at the Roy J. Carver Biotechnology Center (Urbana, IL, USA), as reported by Shao et al. [14]. Briefly, samples were extracted twice with 500 µL of hexane at room temperature and analyzed with a gas chromatography-mass spectrometry system (Agilent Inc., Palo Alto, CA, USA) consisting of an Agilent 7890B gas chromatograph and an Agilent 5977A MSD. Target peaks were evaluated using AMDIS v2.71 and Mass Hunter Qualitative Analysis B.08.00 (Agilent Inc., Palo Alto, CA, USA) software.

The composition analysis of the milk samples was conducted by Dairy Lab Service Inc. (Dubuque, IA, USA). Milk fatty acid profile analysis was conducted by Cumberland Valley Analytical Service Inc. (CVAS; Waynesboro, PA, USA). The procedure of milk fatty acid profile analysis was previously described by Shao et al. [14].

Pasture samples from each group during the supplementation period were composited into one sample, while supplement samples for each ingredient were composited for the gestation and post-calving periods. Composited samples that were going to be analyzed for fatty acid profile were freeze dried, while the remaining samples were oven dried under 55 °C for at least 3 days prior to proximate analysis. All dried samples were ground through a 1 mm screen using a Wiley mill (Arthur, H. Thomas, Philadelphia, PA, USA). Ground samples were analyzed for DM (105 °C oven), crude protein (Leco TruMac, LECO Corporation, St. Joseph, MI, USA), crude fat using an Ankom XT10 Fat Extractor (Ankom Technology, Macedon, NY), NDF, and ADF using an Ankom 200 Fiber Analyzer (Ankom Technology, Macedon, NY, USA). Feed sample fatty acid profile analysis was conducted by CVAS using the method of Sukhija and Palmquist [18] and modified as reported by Shao et al. [14].

The extraction of RNA from biopsy samples, complementary DNA synthesis, and quantitative PCR (qPCR) for relative mRNA expression were conducted, as described by Shao et al. [14]. The primers used for qPCR are listed in Table 4. The data were processed and calculated using QuantStudio Real-Time PCR Software (version 1.3, Thermo Fisher Scientific, Waltham, MA, USA). The target genes were chosen based on their involvement in myogenesis and adipogenesis during growth and development. The final data were normalized using the geometric mean of internal control genes: GAPDH, ACTB, and RPLP0 for muscle samples, ACTB, BRPS2, and SLC35BC for adipose tissue samples. The amplification efficiency of the internal control genes and target genes ranged from 92.5% to 104.6%.

### 2.4. Statistical Analysis

The cow grazing group was considered the experimental unit for all response variables. Outliers were checked by using the Proc Reg procedure of SAS (version 9.4; SAS Institute Inc., Cary, NC, USA), removing data with a studentized t greater than 3 prior to analysis. The MIXED procedure of SAS was used for all response variables except pregnancy rates. A random statement of cow grazing group nested within treatment was included in all analyses when individual animal observational data were used. The model for cow BW and BCS included treatment as the fixed effect and cow age as a covariate. The model for cow milk production and composition included treatment as the fixed effect, and cow milk expected progeny difference (EPD), age, and day postpartum as covariates. The model for relative concentrations of calf plasma fatty acid and cow milk fatty acid profile included treatment as a fixed effect, while day postpartum was included as a covariate for plasma fatty acid. The model for steer growth performance included treatment as a fixed effect and the corresponding EPD as a covariate. Relative mRNA abundance data that were not normally distributed (determined by plotting residuals) were transformed with a logarithmic function (LOG) or a Box-Cox family of power transformations to improve the normality of residuals. The model for relative mRNA expression included treatment, time, and interaction between treatment and time as fixed effects, while sire was included as a random effect. When the treatment effect was detected, all pairwise comparisons were conducted using the PDIFF of SAS at the treatment level. Repeated measure analysis of SAS was used for the analysis of cow plasma fatty acids. Relative concentrations of fatty acids at the initiation of supplementation were tested as a covariate for the analysis of relative concentrations of cow plasma fatty acids, and removed because of non-significance. The model included treatment, time, and the interaction between treatment and time as fixed effects. A simple structure was used as the covariance structure based on the Akaike information criterion. The GLIMMIX procedure of SAS was used for the analysis of pregnancy rates, with treatment as a fixed effect and age as a covariate in the model. Significance was declared at *p* ≤ 0.05, and tendencies were declared from 0.05 < *p* ≤ 0.10.

## 3. Results and Discussion

### 3.1. Cow Parameters

Total supplementation length, supplementation length during gestation, and supplementation length postpartum were not different (*p* ≥ 0.43) among treatments. Relative concentrations of cow plasma fatty acids are presented in Table 5. There tended to be treatment × time interactions (*p* ≤ 0.09) for relative plasma concentrations of C16:0 and C16:1c9 of cows during the supplementation period. Relative concentrations of C16:0 and C16:1c9 of cows from the CON and PUFA groups tended to increase to a greater extent from mid-supplementation to calving compared to those of SFA/MFUA cows. In spring-calving beef cows, late gestation soybean oil supplementation increased C16:0 and decreased C16:1c9 in plasma at calving compared to saturated fat [33]. Under the same spring-calving system, late gestation supplementation with n-3 and n-6 polyunsaturated fatty acids led to decreased plasma C16:0 at calving compared to saturated and monounsaturated fatty acid supplementation [13]. No effects of treatment or treatment × time interaction on concentration of C16:0 or C16:1c9 were reported [14] when cows were supplemented with 80 g DM/cow/d Strata and 80 g DM/cow/d Prequel during late gestation compared with the treatment had the same amount of EnerGII in SFA/MUFA of the current study. Because C16:0 is a common end product of fatty acid synthesis [34] and SFA/MUFA cows had the greatest dietary intake of C16:0, the results might indicate greater fatty acid synthesis in CON and PUFA cows compared to SFA/MUFA at calving. Coleman et al. [35] reported that EPA/DHA supplemented ewes had a greater mRNA concentration of fatty acid synthase (FASN) in subcutaneous adipose tissue, which might support the assumption of greater fatty acid synthesis in PUFA cows in the current study. In addition, a negative energy balance during mid-supplementation to calving, which was indicated by decreased BCS (Table 6), might have led to the mobilization of previously reserved fatty acids. If cows were mobilizing stored fatty acids, the cow plasma fatty acid profile would not have been reflective of dietary fatty acid intake at the same time.

The relative concentrations of C18:2n-6 (Linoleic acid; LA) of SFA/MUFA and PUFA cows were greater (*p* = 0.01) during the supplementation period compared to CON cows but were not different from each other. Consistent with the current study, it was reported that there was no difference in the concentration of C18:2n-6 when ruminant dams were supplemented with polyunsaturated fatty acids during late gestation compared with saturated and monounsaturated fatty acids [14,35]. At mid-supplementation and calving, DHA was detectable in all PUFA groups but only in one group from CON and one group from SFA/MUFA at calving. Consistent with the current study, polyunsaturated fatty acid supplementation increased the plasma DHA concentration compared to saturated and monounsaturated supplementation [13,14].

Cow BW or BCS from experiment initiation through weaning was not different (*p* ≥ 0.13; Table 6) among treatments. Gestation length of the cows from the different treatments was not different (*p* = 0.63; Table 7). Consistent with the current study, previous studies that also supplemented isocaloric fat supplements [13,14,36] reported that maternal fatty acid supplementation during late gestation did not affect cow BW or BCS. In the current study, the BCS of the cows from all treatments decreased from mid-supplementation to calving, which might indicate a negative energy balance during the last 42 d of gestation. Compared to 131 mm of accumulated precipitation for June, July, August, and September in Shao et al. [14], the accumulated precipitation during the late gestation period in the current study was 59 mm, which might negatively impact forage quality. In addition, September is when endophyte-infected tall fescue has a peak concentration of ergot alkaloid [37], which might have led to fescue toxicity for the pregnant cows, causing decreased BCS in the current study.

Milk production and milk composition at WSW did not differ (*p* ≥ 0.12; Table 7) among treatments. Consistent with the current study, it was also reported that late gestation supplementation with polyunsaturated fatty acids had no effects on the milk yield or composition of cows [14] or ewes [35]. For the milk fatty acid profile, there was a treatment effect (*p* = 0.05; Table 8) on the concentrations of C15:0 and total n-3 fatty acids; PUFA cows had greater concentrations than CON cows, while SFA/MUFA cows were intermediate and not different from the other treatments. Concentration of DHA was lower than the detectable level in CON cows’ milk but ranged from 0.01–0.03 g/100 g fatty acid in 4 cows from SFA/MUFA groups and from 0.01–0.04 g/100 g fatty acid in 7 cows from PUFA groups. Coleman et al. [35] reported that EPA + DHA supplementation during late gestation tended to increase ewe milk C15:0 at 30 d postpartum compared to supplementation with palmitic and oleic acids. Pentadecanoic acid (C15:0) originates from rumen microbial fermentation [38]. The greater concentration of milk produced by PUFA cows might indicate greater microbial fermentation at the time of WSW. Since all fatty acids with 18 or more carbons in ruminant milk are derived from either absorption or reserves [39], a greater concentration of total n-3 fatty acids in PUFA cows indicates greater n-3 fatty acid reserves, for example, DHA and C18:3n-3, from the supplementation period compared to CON cows.

Although pregnancy rates for AI and overall were not different (*p* ≥ 0.88; Table 8) among treatments, the authors acknowledge that the current study is not powered for reproductive performance. Importantly, sexed semen was used for AI and may explain the relatively low AI pregnancy rates compared to the previous year [14]. Essential fatty acids could have an impact on the balance of reproductive hormones [40,41]. Therefore, improvements in reproductive performance were commonly reported when fat supplementation was maintained or took place postpartum [42,43,44]. However, results on the effects of late gestation fat supplementation on cow reproductive performance are inconsistent. Improved reproductive performance was reported when forage availability was limited for heifers supplemented with late gestation fat [36]. Supplementing isocaloric fat during late gestation was also reported to have no effects on reproductive performance [14,45], but neither study was likely adequately powered for reproductive parameters. Future studies with an adequate number of animals to investigate the effects of late gestation fatty acid supplementation on cow reproductive performance are needed.

### 3.2. Calf Performance

The plasma fatty acid profile in offspring reflects maternal supplementation during gestation [46] and colostrum/milk consumption [47]. There was a treatment effect (*p* ≤ 0.05; Table 9) detected for steer plasma concentrations of C17:0, C20:0, C20:3n-6, and C20:5n-3 at birth: PUFA steers had greater concentrations than SFA/MUFA, while CON steers were intermediate and not different from the other treatments. The concentration of C20:4n-6 from PUFA steers were greater (*p* = 0.02) than CON and SFA/MUFA at birth. The concentration of C18:2n-6 in PUFA steer plasma was greater (*p* = 0.04) than CON, with SFA/MUFA steers being intermediate and not different from other treatments. Since the ratio of Prequel and Strata (n-6 to n-3) in the PUFA treatment was increased in the current study compared to Shao et al. [14], different changes in steer calf plasma fatty acid concentrations were observed. The current study showed that offspring plasma fatty acid concentrations were influenced more significantly by maternal supplementation compared to dams’ plasma fatty acid concentrations. Since plasma samples were collected from steer calves after milk consumption in the current study, we could not determine whether changes in fatty acids were due to placenta transfer or milk fatty acid consumption in the first few days of their lives. 

The growth performance of the steer progeny during the pre-weaning and backgrounding periods is presented in Table 10. Body weights of the steer progeny at birth, weaning, or the end of the backgrounding phase were not different (*p* ≥ 0.19) among treatments, which led to no difference (*p* = 0.21) in pre-weaning or backgrounding phase ADG. The weaning age of the steers was not different (*p* = 0.63). Shao et al. [14] reported a greater weaning BW of the steers from dams supplemented with saturated and monounsaturated fatty acids compared with polyunsaturated fatty acids. In the current study, the inclusion of Prequel increased with greater n-6 fatty acid inclusion in PUFA. No differences in pre-weaning growth performance are consistent with fatty acid supplementation studies conducted under spring-calving systems [13,33]. In addition, supplementation of EPA + DHA to ewes during late gestation did not affect lamb performance or metabolism through weaning [48]. Bellows et al. [36] reported that fat supplementation to first-calf heifers during the last 65 d of gestation resulted in a greater fall in calf birth BW compared to isocaloric non-fat supplementation. However, fat supplementation did not affect birth BW compared to non-fat supplementation in the current study. Therefore, neither fat supplementation nor different fatty acid profiles had effects on the pre-weaning growth performance of the steer progeny.

### 3.3. Relative mRNA Expression

The relative mRNA expression of myogenic and adipogenic genes in the longissimus muscle of the steers at birth and weaning are presented in Table 11. There were treatment × time interactions (*p* ≤ 0.01) observed for the expression of Paired box protein 7 (PAX7). The mRNA expression of PAX7 decreased to a greater extent from birth to weaning for SFA/MUFA and PUFA steers compared to CON. It was reported [49] that Myogenic factor 5 (MYF5) is important for the activation of other myogenic regulatory factors, such as Myogenic differentiation 1 (MYOD1) and Myogenin (MYOG). The transcription of MYF5 is dependent on the levels of PAX7 [50,51]. Satellite cell activation is important for postpartum muscle growth, and PAX7 is essential for the expansion and differentiation of muscle satellite cells during neonatal and adult muscle growth [52]. Greater mRNA expression of PAX7 in steers born from fatty acid-supplemented dams indicates that maternal fatty acid supplementation during late gestation might lead to greater satellite cell activation. However, other myogenic genes MYF5, MYOG, or MYOD1 were not affected by maternal supplementation, which was consistent with steer growth performance during the pre-weaning period not differing among treatments. There was a treatment effect (*p* = 0.01) on the mRNA expression of MYH7, where the expression during the pre-weaning period was greater in PUFA steers compared to CON and SFA/MUFA. In a previous study [14], where greater n-3 inclusion was used for polyunsaturated fatty acid supplementation treatment, mRNA expression of Longissimus muscle MYH7 in steers from dams supplemented with polyunsaturated fatty acids increased to a greater extent from birth to weaning. In muscle, MYH7 is associated with myosin heavy chain I, which is a slow isoform [53]. The results might indicate that muscle fiber type development was sensitive to the maternal dietary fatty acid profile during late gestation.

There was a treatment × time interaction for the mRNA expression of AGPAT1 and CPT1 (*p* ≤ 0.02) during the pre-weaning period; expression of CON steers increased to a greater extent from birth to weaning compared to SFA/MUFA and PUFA. The mRNA expression of CEBPB tended (*p* = 0.08) to increase to a greater extent from birth to weaning for CON and SFA/MUFA steers compared to PUFA. Treatment tended (*p* = 0.10) to affect the mRNA expression of PPARG, with fatty acid supplementation (SFA/MUFA and PUFA) having greater PPARG mRNA expression compared to non-fat supplementation CON during the pre-weaning period. Acyl-glycerol phosphate acyltransferase 1 (AGPAT1) is one of the critical fatty acid esterification genes and showed a strong positive correlation with intramuscular fat content [43]. Carnitine palmitoyltransferase I is part of the mitochondrial transportation of long-chain fatty acids and is a key enzyme in the regulation of the oxidation of long-chain fatty acids [54]. The essential regulatory genes of adipogenesis include zinc finger protein 423 (ZFP423), CEBPA, CEBPB, and PPARG [11]. It is well recognized that PPARG is essential and indispensable for adipogenesis [55], and its functioning is regulated by CEBPs [56]. Consistent with the current study, Brandão et al. [33] reported that the mRNA expression of PPARG at birth was greater in the longissimus muscles of offspring from dams supplemented with Ca salts of soybean oil. The data in the current study indicate that fatty acid supplementation during late gestation might lead to greater adipogenesis and modified lipid metabolism in the longissimus muscle of the steer progeny during the pre-weaning period, especially in the neonatal stage. Further research is needed to investigate whether the increased mRNA expression of adipogenic genes in fatty acid supplementation leads to greater marbling deposition during the finishing phase.

The relative mRNA expression of adipogenic genes in the subcutaneous adipose tissue of the steers at birth and weaning are presented in Table 12. The mRNA expression of ZFP423 tended (*p* = 0.07) to increase to a greater extent from birth to weaning for CON steers compared to SFA/MUFA and PUFA. The mRNA expression of CEBPB tended (*p* = 0.07) to decrease to a greater extent from birth to weaning for PUFA steers compared to CON and SFA/MUFA. Subcutaneous adipose tissue negatively impacts feed efficiency and carcass values [57]. Zinc finger protein 423 is important for the adipogenic commitment of progenitor cells [58]. The trend for a greater increase of mRNA expression of ZFP423 in CON from birth to weaning could lead to greater 12th rib fat back thickness in later stages of growth and production. Previous work [14] reported a treatment × time interaction for the expression of CEBPB during the pre-weaning period, as the mRNA expression of CEBPB was greater in CON at birth and then decreased to the same level as the polyunsaturated fatty acid supplementation at weaning. There are limited studies investigating the mRNA expression of adipogenic genes in subcutaneous adipose tissue for maternal fatty acid supplementation. The reason why there are different expression results for CEBPB between the previous [14] and the current study could be due to the different fatty acid profiles in the treatments. Greater n-6 fatty acids, such as C18:2n-6, from the PUFA treatment in the current study might lead to greater pro-adipogenic fetal programming effects. However, more research on maternal fat supplementation affecting subcutaneous adipose gene expression is needed to validate the current findings and investigate further effects on finishing phase growth performance.

## 4. Conclusions

Late gestation supplementation of fatty acids or different fatty acid profiles to fall-calving beef cows grazing endophyte-infected tall fescue modified plasma fatty acid profile of cow and steer progeny at birth. However, supplementation did not affect cow BW or BCS. Cow rebreeding pregnancy rates were also not different between treatments; however, this experiment was not powered for reproductive parameters. Myogenic and adipogenic gene mRNA expression in the longissimus muscle and subcutaneous adipose tissue was modified by maternal fatty acid supplementation and fatty acid profile. However, the modification of mRNA expression did not translate into improved steer growth performance during the pre-weaning or backgrounding periods. This experiment was conducted with fall-calving cows grazing on an endophyte-infected tall fescue. Future work exploring other forage systems, calving seasons, and management practices is needed to fully understand the potential of fatty acid supplementation during gestation.

## Figures and Tables

**Table 1 animals-13-00437-t001:** Supplement composition and nutrient intake from treatment supplements containing soybean hulls mixed with whole-shelled corn (CON), Ca salts of saturated/monounsaturated fatty acids (SFA/MUFA), or Ca salts of polyunsaturated fatty acids (PUFA) ^1^.

Item (Dry Matter Basis)	CON	SFA/MUFA	PUFA
Ingredients, kg/cow/d			
Soybean hull	0.16	0.77	0.77
Whole-shelled corn	0.91	-	-
EnerGII	-	0.155	-
Prequel	-	-	0.12
Strata	-	-	0.04
Macronutrient intake, kg/cow/d			
Dry matter	1.07	0.93	0.93
Crude protein	0.08	0.08	0.08
Total fat	0.05	0.16	0.15
Total digestible nutrients ^2^	0.90	0.90	0.89
Fatty acid intake, g/d			
C16:0	4.0	55.6	21.7
C16:1c9	0.1	0.3	3.4
C18:0	0.6	5.8	5.6
C18:1c9	7.1	42.9	24.9
C18:2n-6	13.4	13.7	45.8
C18:3n-3	0.6	1.5	2.5
C20:5n-3	-	-	2.8
C22:6n-3	-	-	1.9

^1^ CON = 0.16 kg DM soybean hulls/cow/d mixed with 0.91 kg DM whole-shelled corn/cow/d. SFA/MUFA = 0.77 kg DM soybean hulls/cow/d mixed with 0.155 kg DM EnerGII (Virtus Nutrition LLC, Corcoran, CA, USA)/cow/d; PUFA = 0.77 kg DM soybean hulls/cow/d mixed with 0.04 kg DM Strata + 0.12 kg DM Prequel (Virtus Nutrition LLC, Corcoran, CA, USA)/cow/d. ^2^ Calculated with book values for soybean hulls and whole-shelled corn, and product label values for EnerGII, Prequel, and Strata.

**Table 2 animals-13-00437-t002:** Nutritional and fatty acid profiles of ingredients fed to cows for the last 82 ± 5 d of gestation ^1^.

Item (% of Dry Matter)	Forage	Soybean Hull	Whole-Shelled Corn	EnerGII	Prequel	Strata
Dry matter, %	30.2	86.9	87.2	96.5	95.6	96.3
Crude protein, %	14.3	10.0	7.2	-	-	-
Neutral detergent fiber, %	52.5	65.5	7.4	-	-	-
Acid detergent fiber, %	28.2	48.3	2.6	-	-	-
Total fat, %	2.9	3.0	3.6	83.9	78.2	78.2
Total fatty acid, %	1.1	1.7	2.6	71.1	68.2	59.7
Fatty acid profile, % of total fatty acid						
C16:0	21.8	18.6	14.6	48.3	15.3	28.5
C16:1c9	2.3	0.8	0.2	0.2	0.6	11.6
C18:0	2.5	6.5	1.9	4.5	4.2	5.8
C18:1c9	5.8	17.2	27.9	36.9	26.9	2.8
C18:2n-6	21.6	40.6	52.1	7.7	48.2	4.7
C18:3n-3	37.4	9.6	1.4	0.2	1.2	1.2
C20:5n-3	0	0	0	0	0	11.6
C22:6n-3	0	0	0	0	0	7.8

^1^ EnerGII, Prequel, and Strata were from Virtus Nutrition LLC, Corcoran, CA, USA.

**Table 3 animals-13-00437-t003:** Nutrient composition of the backgrounding diet for steers.

Item	Common Diet
Ingredients, % DM basis	
Whole-shelled corn	50.5
Corn distillers grains	34.9
Co-product balancer ^1^	4.4
Hay	10.2
Analyzed nutrient content, % DM	
DM	86.3
CP	14.8
NDF	35.8
ADF	12.1
Crude fat	4.2

^1^ Co-product balancer contained 25% crude protein, 1.5% crude fat, 8.0% crude fiber, 14.0% Ca, 0.1% P, 0.5% Mg, 0.1% K, 3.5% salt, 300 mg/kg Cu, 3.0 mg/kg Se, 1500 mg/kg Zn, 52.9 KIU/kg vitamin A, 5.3 KIU/kg vitamin D3, 55 IU/kg vitamin E.

**Table 4 animals-13-00437-t004:** GenBank accession number, sequence, and amplicon size of primers used for mRNA expression by qPCR.

Accession Number	Gene ^1^	Direction	Primer Sequences (5′ to 3′)	Amplicon Length (bp)
NM_001034034.2	*GAPDH* [19]	F78	AAGGTCGGAGTGAACGGATTC	98
		R175	AAGGGGTCATTGATGGCGAC	
NM_173979.3	*ACTB* [20]	F862	GCCCTGAGGCTCTCTTCCA	100
		R961	CGGATGTCGACGTCACACTT	
NM_001012682.1	*RPLP0* [21]	F657	CAACCCTGAAGTGCTTGACAT	227
		R883	AGGCAGATGGATCAGCCA	
NM_001033613	*BRPS2* [22]	F320	GGAGCATCCCTGAAGGATGA	101
		R420	TCCCCGATAGCAACAAACG	
BC103464	*SLC35B2* [23]	F894	ACATTGCTTTCGACAGCTTCAC	95
		R988	GAAGAGATTGACCCCAAACATCA	
NM_174116.1	*MYF5* [24]	F892	CCTCTAGTTCCAGGCTCATCTA	90
		R981	ACCTCCTTCCTCCTGTGTAATA	
NM_001111325	*MYOG* [24]	F222	GGCGTGTAAGGTGTGTAAG	85
		R306	CTTCTTGAGTCTGCGCTTCT	
NM_001040478.2	*MYOD1* [14]	F827	AACTGTTCCGACGGCATGAT	105
		R931	CCGGGGTTCGTTGGGC	
XM_015460690.2	*PAX7* [14]	F435	AGATCGAGGAGTACAAGAGGGA	112
		R545	ATCGAACTCACTGAGGGCACG	
NM_174727.1	*MYH7* [14]	F2160	TTCCGGCAGAGGTATCGAAT	128
		R2287	TGGCCGAACTTATACTGGTTGTG	
NM_001046113	*MEF2C* [24]	F703	CCTGATGCAGACGATTCAGTAG	123
		R825	AAAGTTGGGAGGTGGAACAG	
NM_001101893.1	*ZFP423* [25]	F240	GGATTCCTCCGTGACAGCA	120
		R359	TCGTCCTCATTCCTCTCCTCT	
NM_181024.2	*PPARG* [26]	F135	CCAAATATCGGTGGGAGTCG	101
		R235	ACAGCGAAGGGCTCACTCTC	
NM_176784.2	*CEBPA* [14]	F385	TTCAACGACGAGTTCCTGGC	107
		R491	CCCGGGTAGTCAAAGTCGTT	
NM_176788.1	*CEBPB* [25]	F333	CGGGCAGCACCACGACTTCC	106
		R438	CCCCAGTCGGCCCAGACTCA	
NM_174314.2	*FABP4* [27]	F401	TGGTGCTGGAATGTGTCATGA	101
		R501	TGGAGTTCGATGCAAACGTC	
NM_177945	*PPARGC1A* [28]	F2001	GTACCAGCACGAAAGGCTCAA	120
		R2120	ATCACACGGCGCTCTTCAA	
NM_177518	*AGPAT1* [29]	F563	TGCCATCAGTGTCATGTCTG	86
		R648	GGTTTCTCGTGCCCTCAG	
CR552737	*FASN* [30]	F6383	ACCTCGTGAAGGCTGTGACTCA	92
		R6474	TGAGTCGAGGCCAAGGTCTGGAA	
NM_001113302.1	*SREBP1* [31]	F2858	TACCTGCAGCTTCTCCATCA	145
		R3002	CACCAATGGGTACAGCCTCT	
NM_173959.4	*SCD* [32]	F809	TCCTGTTGTTGTGCTTCATCC	101
		R909	GGCATAACGGAATAAGGTGGC	
NM_174224.2	*ACACA* [14]	F182	TGTGAAGTATCCTTCTGGAGGT	99
		R280	CTTCCAAAAAGAACTCAGAGACC	

^1^*MYH7* Myosin heavy chain 7, *PAX7* Paired box protein 7, *MYF5* Myogenic factor 5, *MYOD1* Myogenic differentiation 1, *MYOG* Myogenin, *MEF2C* Myocyte enhancer factor 2C, *CEBPA* CCAAT enhancer binding protein alpha, *CEBPB* CCAAT enhancer binding protein beta, *ZFP423* Zinc finger protein 423, *FABP4* Fatty acid binding protein 4, *PPARG* Peroxisome proliferator activated receptor gamma, *GPAT1* Glycerol-3-phosphate acyltransferase 1, *PPARGC1A* PPARG coactivator 1 alpha, *SCD* Stearoyl-CoA desaturase, *AGPAT1* Acyl-glycerol phosphate acyltransferase 1, *FASN* Fatty acid synthase, *SREBP1* Sterol regulatory element binding transcription factor 1, *ACACA* Acetyl-CoA carboxylase alpha.

**Table 5 animals-13-00437-t005:** Effects of supplementation for the last 82 ± 5 d of gestation of soybean hulls mixed with either whole-shelled corn (CRN), Ca salts of saturated/monounsaturated fatty acids (SFA/MUFA), or Ca salts of polyunsaturated fatty acids (PUFA2) on the relative concentrations of selected plasma fatty acids in cows ^1^.

Item	Mid-Sup ^2^			At-Calving ^3^			SEM	*p*-Value ^4^		
	CON ^5^	SFA/ MUFA ^6^	PUFA ^7^	CON	SFA/MUFA	PUFA		Trt	Time	Trt x Time
Groups, n	4	4	4	4	4	4				
C15:0	44	51	44	31	27	37	4	0.76	<0.01	0.17
C16:0	1008	1353	1078	1373	1274	1589	121.8	0.47	0.03	0.09
C16:1c9	17.8	16.4	7.1	46.4	31.8	45.9	4.73	0.29	<0.01	0.09
C17:0	68	78	77	49	42	58	5.2	0.22	<0.01	0.26
C18:0	1420	1787	1468	1303	1219	1465	124.9	0.52	0.05	0.11
C18:1c9	481	669	446	975	774	696	120.6	0.38	0.02	0.31
C18:2n-6	975	1705	1499	779	977	1379	150.3	0.01	0.02	0.14
C18:3n-3	277	316	196	159	169	183	45.7	0.53	0.04	0.35
C20:0	3.2	3.8	3.9	2.4	3.0	4.0	0.57	0.16	0.32	0.69
C20:3n-6	114	128	103	67	56	65	12.1	0.75	<0.01	0.30
C20:4n-6	132	176	141	138	131	169	20.3	0.57	0.83	0.23
C20:5n-3	34	41	49	31	33	46	7.1	0.15	0.47	0.92

^1^ Data are presented as relative concentration per 100 ul plasma to internal standard C23:0; the concentration of the corresponding fatty acid prior to supplementation was included as covariate if significant. ^2^ Mid-supplementation was d -42 of the experiment. ^3^ At-calving plasma samples were collected from dams 4 ± 2.2 d post-calving. ^4^ Trt = treatment effect; Trt × Time = interaction between treatment and time. ^5^ CON = 0.16 kg DM soybean hulls/cow/d mixed with 0.91 kg DM whole-shelled corn/cow/d. ^6^ SFA/MUFA = 0.77 kg DM soybean hulls/cow/d mixed with 0.155 kg DM EnerGII (Virtus Nutrition LLC, Corcoran, CA, USA)/cow/d. ^7^ PUFA = 0.77 kg DM soybean hulls/cow/d mixed with 0.04 kg DM Strata + 0.12 kg DM Prequel (Virtus Nutrition LLC, Corcoran, CA, USA)/cow/d.

**Table 6 animals-13-00437-t006:** Effects of supplementation for the last 82 ± 5 d of gestation of soybean hulls mixed with either whole-shelled corn (CON), Ca salts of saturated/monounsaturated fatty acids (SFA/MUFA), or Ca salts of polyunsaturated fatty acids (PUFA) on cow body weight and body condition score ^1^.

Item ^1^	CON ^2^	SFA/MUFA ^3^	PUFA ^4^	SEM	*p*-Value
Cow BW, kg					
Initial (d -82)	611	613	608	13.7	0.93
Mid supplementation (d -42)	662	665	661	14.4	0.96
Calving (4 ± 2 d post-calving)	628	637	623	14.8	0.60
WSW (d 68)	622	636	626	15.9	0.66
AI (d 85)	629	649	638	16.5	0.47
AI-conception check (d 120)	567	590	601	15.9	0.13
Weaning (d 174)	563	578	569	16.1	0.64
Cow BCS					
Initial	6.1	6.1	6.0	0.19	0.98
Mid supplementation	6.2	6.6	6.2	0.24	0.19
Calving	5.4	5.4	5.3	0.18	0.55
WSW	5.3	5.6	5.2	0.15	0.11
AI	5.1	5.2	4.9	0.22	0.45
AI-conception check	5.0	5.0	5.0	0.04	0.29
Weaning	5.0	5.1	5.0	0.07	0.64

^1^ BW = body weight; WSW = weigh-suckle-weigh; AI = artificial insemination; BCS = body condition score. ^2^ CON = 0.16 kg DM soybean hulls/cow/d mixed with 0.91 kg DM whole-shelled corn/cow/d. ^3^ SFA/MUFA = 0.77 kg DM soybean hulls/cow/d mixed with 0.155 kg DM EnerGII (Virtus Nutrition LLC, Corcoran, CA, USA)/cow/d. ^4^ PUFA = 0.77 kg DM soybean hulls/cow/d mixed with 0.04 kg DM Strata + 0.12 kg DM Prequel (Virtus Nutrition LLC, Corcoran, CA, USA)/cow/d.

**Table 7 animals-13-00437-t007:** Effects of supplementation for the last 82 ± 5 d of gestation of soybean hulls mixed with either whole-shelled corn (CON), Ca salts of saturated/monounsaturated fatty acids (SFA/MUFA), or Ca salts of polyunsaturated fatty acids (PUFA) on milk production and cow reproductive performance.

Item ^1^	CON ^2^	SFA/MUFA ^3^	PUFA ^4^	SEM	*p*-Value
Gestation length, d	279	279	278	1.3	0.63
Milk production^1^ (kg/d)	9.3	8.1	8.7	1.78	0.80
Composition					
Fat, %	3.2	2.5	3.1	0.47	0.28
Protein, %	2.7	2.8	3.0	0.12	0.12
Lactose, %	4.9	4.8	4.6	0.18	0.28
Other solids, %	6.0	5.9	5.7	0.16	0.29
Total solids, %	11.9	11.2	11.8	0.42	0.24
MUN, mg/dL	5.0	4.1	5.1	0.75	0.31
AI pregnancy, %	31.6	34.4	29.0	-	0.96
Overall pregnancy, %	99.3	98.5	98.9	-	0.88

^1^ Milk production was determined via weigh-suckle-weigh technique at 68 ± 5 d postpartum on a subset of 77 pairs (4–8 pairs/group); Milk composition conducted on a subset of cows (4 cows/group; 47 cows in total) at 68 ± 5 d postpartum; MUN = milk urea nitrogen; AI = artificial insemination. ^2^ CON = 0.16 kg DM soybean hulls/cow/d mixed with 0.91 kg DM whole-shelled corn/cow/d. ^3^ SFA/MUFA = 0.77 kg DM soybean hulls/cow/d mixed with 0.155 kg DM EnerGII (Virtus Nutrition LLC, Corcoran, CA, USA)/cow/d. ^4^ PUFA = 0.77 kg DM soybean hulls/cow/d mixed with 0.04 kg DM Strata + 0.12 kg DM Prequel (Virtus Nutrition LLC, Corcoran, CA, USA)/cow/d

**Table 8 animals-13-00437-t008:** Effects of supplementation for the last 82 ± 5 d of gestation of soybean hulls mixed with either whole-shelled corn (CON), Ca salts of saturated/monounsaturated fatty acids (SFA/MUFA), or Ca salts of polyunsaturated fatty acids (PUFA) on milk fatty acid profile ^1^.

Item (g/100 g Fatty Acid)	CON ^2^	SFA/MUFA ^3^	PUFA ^4^	SEM	*p*-Value
Groups, n	4	4	4		
C4:0	4.1	4.1	4.1	0.15	0.99
C6:0	1.8	1.8	1.8	0.06	0.51
C8:0 ^5^	1.0	1.1	1.0	-	0.31
C10:0	2.2	2.4	2.2	0.13	0.41
C12:0	2.5	2.7	2.5	0.16	0.45
C14:0	8.9	9.4	9.1	0.44	0.54
C15:0	1.1 ^b^	1.2 ^ab^	1.3 ^a^	0.05	0.05
C16:0	27	26	26	0.8	0.76
C16:1 c9	2.1	1.9	2.0	0.08	0.07
C17:0	0.9	0.9	1.0	0.03	0.08
C18:0	10.1	10.4	10.6	0.67	0.76
C18:1 t6, 8	0.22	0.24	0.24	0.015	0.47
C18:1 t9	0.19	0.19	0.18	0.011	0.78
C18:1 t11	2.2	2.3	2.3	0.14	0.77
C18:1 t12	0.19	0.19	0.19	0.012	0.86
C18:1 c9	24	23	23	0.9	0.34
C18:1 c11	0.58	0.52	0.57	0.033	0.30
C18:1 c12	0.07	0.07	0.07	0.005	0.90
C18:1 t16	0.16	0.17	0.17	0.010	0.43
C18:2 c9, c12	1.9	2.0	2.0	0.10	0.37
C20:0 ^5^	0.18	0.19	0.20	-	0.30
C20:1 c11 ^5^	0.05	0.04	0.05	-	0.28
C18:3 c9, c12, c15	0.54	0.58	0.64	0.037	0.07
CLA c9, t11	1.08	1.08	1.05	0.063	0.80
C22:0	0.07	0.08	0.09	0.008	0.06
C20:3 n6 ^5^	0.07	0.08	0.08	-	0.29
C20:4 n6 ^5^	0.14	0.14	0.14	-	0.98
C20:5 n3	0.05	0.05	0.05	0.004	0.23
C22:5 n3	0.11	0.12	0.13	0.010	0.41
Total n-3	0.70 ^b^	0.75 ^ab^	0.83 ^a^	0.008	0.05
Total n-6	2.1	2.2	2.3	0.12	0.39

^1^ Milk fatty acid profile analysis was conducted on a subset of cows (4 cows/group; 47 cows in total) at 68 ± 5 d postpartum. ^2^ CON = 0.16 kg DM soybean hulls/cow/d mixed with 0.91 kg DM whole-shelled corn/cow/d. ^3^ SFA/MUFA = 0.77 kg DM soybean hulls/cow/d mixed with 0.155 kg DM EnerGII (Virtus Nutrition LLC, Corcoran, CA, USA)/cow/d. ^4^ PUFA = 0.77 kg DM soybean hulls/cow/d mixed with 0.04 kg DM Strata + 0.12 kg DM Prequel (Virtus Nutrition LLC, Corcoran, CA, USA)/cow/d. ^5^ Data were transformed for statistical analysis because of non-normal distribution; Means are presented after back-transformation. Means within a row with different superscript letters differ (*p* < 0.05).

**Table 9 animals-13-00437-t009:** Effects of supplementation of soybean hulls mixed with either whole-shelled corn (CON), Ca salts of saturated/monounsaturated fatty acids (SFA/MUFA), or Ca salts of polyunsaturated fatty acids (PUFA) for the last 82 ± 5 d of gestation on the relative concentrations of selected plasma fatty acids in steers at birth ^1^.

Item ^1^	CON ^2^	SFA/MUFA ^3^	PUFA ^4^	SEM	*p*-Value
Groups, n	4	4	4		
C15:0	20	22	26	3.1	0.38
C16:0	1080	1168	1206	47.6	0.21
C16:1c9	34	24	34	6.1	0.43
C17:0	33 ^ab^	28 ^b^	47 ^a^	4.5	0.04
C18:0	932	976	1071	46.2	0.16
C18:1c9	901	789	783	55.9	0.30
C18:2n-6	624 ^b^	746 ^ab^	990 ^a^	86.1	0.04
C18:3n-3	103	60	113	19.5	0.19
C20:0	3.6 ^ab^	1.9 ^b^	5.9 ^a^	0.74	0.02
C20:3n-6	28 ^ab^	19 ^b^	51 ^a^	7.6	0.05
C20:4n-6	103 ^b^	70 ^b^	171 ^a^	20.3	0.02
C20:5n-3	27 ^ab^	15 ^b^	44 ^a^	5.4	0.02

^1^ Data are presented as relative concentration per 100 µL plasma to internal standard C23:0; at birth plasma samples were collected from steers 4 ± 2.2 d of age. ^2^ CON = 0.16 kg DM soybean hulls/cow/d mixed with 0.91 kg DM whole-shelled corn/cow/d. ^3^ SFA/MUFA = 0.77 kg DM soybean hulls/cow/d mixed with 0.155 kg DM EnerGII (Virtus Nutrition LLC, Corcoran, CA, USA)/cow/d. ^4^ PUFA = 0.77 kg DM soybean hulls/cow/d mixed with 0.04 kg DM Strata + 0.12 kg DM Prequel (Virtus Nutrition LLC, Corcoran, CA, USA)/cow/d. Means within a row with different superscript letters differ (*p* < 0.05).

**Table 10 animals-13-00437-t010:** Effects of supplementation for the last 82 ± 5 d of gestation of soybean hulls mixed with either whole-shelled corn (CON), Ca salts of saturated/monounsaturated fatty acids (SFA/MUFA), or Ca salts of polyunsaturated fatty acids (PUFA) on steer progeny growth performance during pre-weaning and backgrounding periods.

Item ^1^	CON ^2^	SFA/MUFA ^3^	PUFA ^4^	SEM	*p*-Value
Birth BW, kg	33.7	34.1	32.7	0.74	0.19
Weaning BW, kg	183	186	190	4.5	0.34
Pre-weaning ADG, kg/d	0.85	0.87	0.90	0.027	0.21
Weaning age, d	174	173	175	1.4	0.63
Backgrounding BW, kg	244	244	248	6.2	0.76
Backgrounding^5^ ADG, kg/d	1.43	1.40	1.43	0.057	0.85

^1^ BW = body weight; ADG = average daily gain. ^2^ CON = 0.16 kg DM soybean hulls/cow/d mixed with 0.91 kg DM whole-shelled corn/cow/d. ^3^ SFA/MUFA = 0.77 kg DM soybean hulls/cow/d mixed with 0.155 kg DM EnerGII (Virtus Nutrition LLC, Corcoran, CA, USA)/cow/d. ^4^ PUFA = 0.77 kg DM soybean hulls/cow/d mixed with 0.04 kg DM Strata + 0.12 kg DM Prequel (Virtus Nutrition LLC, Corcoran, CA, USA)/cow/d. ^5^ Backgrounding phase was 42 days.

**Table 11 animals-13-00437-t011:** Relative mRNA expression of genes regulating myogenesis and adipogenesis in the longissimus muscle of the steers (4 steers per group) born from dams supplemented for the last 82 ± 5 d of gestation with soybean hulls mixed with either whole-shelled corn (CON), Ca salts of saturated/monounsaturated fatty acids (SFA/MUFA), or Ca salts of polyunsaturated fatty acids (PUFA) ^1^.

Gene ^6^	Birth			Weaning			*p*-Value ^5^		
	CON ^2^	SFA/MUFA ^3^	PUFA ^4^	CON	SFA/MUFA	PUFA	Trt	Time	Trt × Time
*MYOD1*	1.033	1.080	1.191	0.997	0.770	0.891	0.56	0.02	0.36
*MYOG*	0.912	1.020	0.982	0.897	0.940	0.975	0.69	0.68	0.93
*PAX7*	0.992 ^b^	1.232 ^a^	1.230 ^a^	0.910 ^b^	0.745 ^c^	0.753 ^c^	0.98	<0.01	<0.01
*MYF5*	0.992	1.071	1.059	0.834	0.784	0.833	0.89	<0.01	0.61
*MYH7*	0.614	0.597	0.846	1.225	1.193	1.467	0.01	<0.01	0.44
*AGPAT1*	0.786 ^c^	0.904 ^b^	0.989 ^ab^	1.036 ^a^	0.915 ^ab^	0.893 ^b^	0.55	0.08	<0.01
*CPT1*	0.739 ^c^	0.795 ^bc^	1.040 ^a^	0.924 ^ab^	0.814 ^bc^	0.895 ^abc^	0.15	0.53	0.02
*PPARG*	0.112	0.125	0.138	0.213	0.246	0.231	0.10	<0.01	0.36
*ZFP423*	0.641	0.732	0.810	1.072	1.193	1.211	0.12	<0.01	0.68
*CEBPA*	0.076	0.074	0.093	0.411	0.510	0.454	0.47	<0.01	0.28
*CEBPB*	0.057	0.058	0.081	0.297	0.350	0.342	0.12	<0.01	0.08

^1^ Means are back-transformed if transformation was conducted. ^2^ CON = 0.16 kg DM soybean hulls/cow/d mixed with 0.91 kg DM whole-shelled corn/cow/d. ^3^ SFA/MUFA = 0.77 kg DM soybean hulls/cow/d mixed with 0.155 kg DM EnerGII (Virtus Nutrition LLC, Corcoran, CA)/cow/d. ^4^ PUFA2 = 0.77 kg DM soybean hulls/cow/d mixed with 0.04 kg DM Strata + 0.12 kg DM Prequel (Virtus Nutrition LLC, Corcoran, CA, USA)/cow/d. ^5^ Trt = treatment effect; Trt × Time = interaction between treatment and time. ^6^ MYOD1 Myogenic differentiation 1; MYOG Myogenin; PAX7 Paired box protein 7; MYF5 Myogenic factor 5; MYH7 Myosin heavy chain 7; AGPAT1 Acyl-glycerol phosphate acyltransferase 1; CPT1 Carnitine palmitoyltransferase 1; PPARG Peroxisome proliferator activated receptor gamma; ZFP423 Zinc finger protein 423; CEBPA CCAAT enhancer binding protein alpha; CEBPB CCAAT enhancer binding protein beta. Means within a row with different superscript letters differ (*p* < 0.05).

**Table 12 animals-13-00437-t012:** Relative mRNA expression of genes regulating adipogenesis in subcutaneous adipose tissue of the steers (4 steers per group) born from dams supplemented soybean hulls mixed with either whole-shelled corn (CRN), Ca salts of saturated/monounsaturated fatty acids (SFA/MUFA), or Ca salts of polyunsaturated fatty acids (PUFA2) for the last 82 ± 5 d of gestation ^1^.

Gene ^6^	Birth			Weaning			*p*-Value ^5^		
	CON ^2^	SFA/MUFA ^3^	PUFA ^4^	CON	SFA/ MUFA	PUFA	Trt	Time	Trt × Time
*ZFP423*	1.055	1.114	1.134	1.324	1.205	1.073	0.48	0.09	0.07
*ACACA*	1.470	1.537	1.309	0.157	0.250	0.169	0.75	<0.01	0.86
*CEBPA*	0.624	0.692	0.875	0.665	0.615	0.622	0.64	0.16	0.21
*CEBPB*	0.469	0.477	0.619	0.527	0.466	0.489	0.16	0.41	0.07
*PPARG*	1.039	0.964	1.132	0.848	0.794	0.757	0.87	<0.01	0.58
*SCD*	0.634	0.572	0.462	0.082	0.097	0.079	0.72	<0.01	0.78
*FASN*	0.991	0.591	0.672	0.117	0.103	0.131	0.69	<0.01	0.69
*FABP4*	0.974	0.874	1.220	1.044	0.942	0.905	0.61	0.53	0.16

^1^ Means are back-transformed if transformation was conducted. ^2^ CRN = 0.16 kg DM soybean hulls/cow/d mixed with 0.91 kg DM whole-shelled corn/cow/d. ^3^ SFA/MUFA = 0.77 kg DM soybean hulls/cow/d mixed with 0.155 kg DM EnerGII (Virtus Nutrition LLC, Corcoran, CA, USA)/cow/d; ^4^ PUFA2 = 0.77 kg DM soybean hulls/cow/d mixed with 0.04 kg DM Strata + 0.12 kg DM Prequel (Virtus Nutrition LLC, Corcoran, CA, USA)/cow/d. ^5^ Trt = treatment effect; Trt × Time = interaction between treatment and time. ^6^ ZFP423 Zinc finger protein 423; ACACA Acetyl-CoA carboxylase alpha; CEBPA CCAAT enhancer binding protein alpha; CEBPB CCAAT enhancer binding protein beta; PPARG Peroxisome proliferator activated receptor gamma; SCD Stearoyl-CoA desaturase; FASN Fatty acid synthase; FABP4 Fatty acid binding protein 4.

## Data Availability

The raw datasets used and analyzed during the current study are available from the corresponding author on reasonable request.

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
