# Peer review of "Effects of Late Gestation Supplements Differing in Fatty Acid Amount and Profile to Beef Cows on Cow Performance, Steer Progeny Growth Performance through Weaning, and Relative mRNA Expression of Genes Associated with Muscle and Adipose Tissue Development"

_animals, 2023, doi:10.3390/ani13030437_

Round 1

Reviewer 1 Report

1. Please explain how to deal with simultaneous estrus in detail.

2. The somatic cell number is a very important indicator to measure the health status of cows. But there are not the number of somatic cells in Table 7. Please explain the reason.

3. Please explain why was milk production and the milk fat percentage lower in the SFA/MUFA group than in the control group.

 4. If the effects on reproductive performance are to be studied, the time of first estrus and gonadal hormone levels during late gestation and after calving need to be measured and counted. But nothing about it appears in the article.

 5. The contents in Table 11, 12 can be converted into a bar chart.

 6. Where is Table 10?

 7. The number of test animal samples for each group was not available in Table 5, 7, 8, 9, 11 and 12.

 8. Why not use RNA-Seq Transcriptome to analysis to the relative expression of genes associated with muscle and adipose tissue development?

Reviewer 2 Report

General comments:

The manuscript “Effects of late gestation supplements differing in fatty acid amount and profile to beef cows on cow performance, steer progeny growth performance through weaning, and relative mRNA expression of genes associated with muscle and adipose tissue development” is an important and actual topic.

Specific comments:

L10: please report more general benefits due to your results, like economics or for the whole sector

L18: please rewrite the beginning of the abstract, an introduction is needed

L18: more methods are needed and maybe fewer results

L49: please report the reference

L52: report reference

L54: report reference

L57: here I suggest to cite 10.3390/antiox11050827

L62: report reference

LL105-106: please report more information regarding the pasture.

Table 1: How you have measured the fatty acid intake? Please report the method in the m&m section.

Table 2: was the forage checked for quality? please report and cite: 10.1016/j.jevs.2022.103940 to ensure that

L126: please cite: 10.1080/1828051X.2022.2032850 for BCS method

L156: please report the calving characteristics (easy, hard) and the recovery after calving as reported: 10.3168/jds.2020-18867, I suggest also as a citation

L 263: report the test for normal distribution

Table 11: report the meaning of the letters

L538: please report the limitations of the study.

L540: so, it is confirmed or not the literature?

L542: that was new or not?

L546: you repeat the statement before, conclusions need to improve

Round 2

Reviewer 1 Report

Please provide original pictures and data about mRNA expression of genes.

Author Response

mRNA expression results are currently provided in customary format for publication and they are similar to many prior publications (https://doi.org/10.1186/s40104-021-00588-w; https://doi.org/10.3389/fmicb.2016.00701). Pictures of the gel are not generated in qPCR as the florescence is detected by the qPCR machine to generate the CT value that is used in the analysis. These CT values are analyzed as currently described in the methods to generate the results.

Reviewer 2 Report

The authors have satisfactorily responded to all my questions and made the necessary changes to the manuscript. The revised version of the manuscript appears to be good. It looks ready for publication as far as I can tell.

Author Response

Thank you.